# Bruton’s Tyrosine Kinase Inhibitors: Recent Updates

**DOI:** 10.3390/ijms25042208

**Published:** 2024-02-12

**Authors:** Amneh Fares, Carlos Carracedo Uribe, Diana Martinez, Tauseef Rehman, Carlos Silva Rondon, Jose Sandoval-Sus

**Affiliations:** 1Memorial Healthcare System, Pembroke Pines, FL 33021, USA; ccarradedouribe@mhs.net (C.C.U.); dianamartinez@mhs.net (D.M.);; 2Moffitt Malignant Hematology at Memorial Healthcare System, Pembroke Pines, FL 33021, USAjsandovalsus@mhs.net (J.S.-S.)

**Keywords:** Bruton’s kinase inhibitors, ibrutinib, acalabrutinib, zanubrutinib, pirtobrutinib, hematologic malignancies, autoimmune disorders

## Abstract

Bruton’s tyrosine kinase (BTK) inhibitors have revolutionized the landscape for the treatment of hematological malignancies, solid tumors, and, recently, autoimmune disorders. The BTK receptor is expressed in several hematopoietic cells such as macrophages, neutrophils, mast cells, and osteoclasts. Similarly, the BTK receptor is involved in signaling pathways such as chemokine receptor signaling, Toll-like receptor signaling, and Fc receptor signaling. Due to their unique mechanism, these agents provide a diverse utility in a variety of disease states not limited to the field of malignant hematology and are generally well-tolerated.

## 1. Introduction

Bruton’s tyrosine kinase (BTK) plays an active role in B-cell differentiation, proliferation, and survival upon stimulation of multiple signal transduction pathways downstream to the B-cell antigen receptor (BCR). Originally, BTK was thought to be related to the X-linked agammaglobulinemia, an immunodeficiency disorder preventing the transformation of pre-B-cells into mature B-cells. Later on, it was identified that BTK was highly expressed in a wide variety of immune cells with identified molecular effects within the myeloid lineage, which have been linked to the modulation of the tumor microenvironment, making BTK a suitable drug target with promising anti-tumor and anti-inflammatory activity. To date, BTK inhibitors (BTKis) have revolutionized the treatment landscape of multiple hematological malignancies, solid tumors, and, more recently, autoimmune disorders. Recent data have identified an outstanding role for BTKis in the pathogenesis of inflammatory diseases and autoimmune disorders. BTK inhibition propels diverse molecular effects that transcend beyond its classic antitumor role. Both pre-clinical and clinical studies have highlighted BTK as an essential receptor in defining B-cell activation thresholds and the autoreactivity of B-cells through the BCR signaling pathway. Moreover, the lack of well-tolerated therapeutic alternatives for autoimmune disorders and the emerging data of BTKis in autoimmune disorders pose an exciting novel approach to treating patients suffering from detrimental autoimmune disorders, where BTKis have significantly improved outcomes and have presented new hope for a sustainable clinical remission [1,2]. The aim of this review is to provide an overview of BTKis and the most recent data available surrounding BTKis, including newer indications such as autoimmune disorders.

## 2. BCR Signaling Pathway Overview

The BCR signaling pathway is a critical element of the B-cell fate and functionality (Figure 1). The receptor’s antigen binding part is composed of a surface-bound immunoglobulin (Ig), which is non-covalently associated with a heterodimeric signaling co-receptor comprised of Igα and Igβ chains. Antigenic binding to BCR causes aggregation and a signal transduction and phosphorylation cascade that ultimately recruits BTK, a non-receptor tyrosine kinase that plays a significant role in the signal transduction of the BCR and other cell surface receptors in both normal and malignant B-lymphocytes. These various signaling pathway activations ultimately result in B-cell survival, proliferation, and differentiation [3,4].

### 2.1. BTK Receptor

BTK is a cytoplasmatic non-receptor tyrosine kinase belonging to the Tec family of kinases. In humans, Tec family proteins are significantly expressed in hematopoietic tissues and facilitate the first step in antigen receptor signaling. BTK is expressed mainly in B-cells, myeloid cells, and platelets. In contrast, T-lymphocytes and plasma cells have low or undetectable levels of BTK [4,5].

BTK consists of 659 amino acids and five domains, which include the pleckstrin homology (PH) domain, the proline-rich TEC homology (TH) domain, the SRC homology (SH) domains (named SH3 and SH2), and finally, the catalytic domain. The PH domain mediates protein–phospholipid and protein–protein interactions. The TH domain contains a zinc finger motif important for protein activity and stability. Both SH2 and SH3 domains contain the autophosphorylation site Tyr223, whereas the catalytic domain contains two phosphorylation sites (Tyr551 and Cys481) targeted by irreversible inhibitors. As a result, these complex structural characteristics are fundamental for the design of different types of BTKis [1].

### 2.2. BTK Inhibition

In general, BTK inhibition causes a block of different downstream cell signaling pathways strictly related to the development of B-cell malignancies, as well as autoimmune diseases by impairing cell proliferation, migration, and the activation of NF-κB. BTK inhibitors can be grouped into two main types considering their mechanism of action and binding modes—reversible and irreversible. Irreversible inhibitors are characterized by a Michael acceptor moiety able to form a covalent bond with the conserved Cys481 residue in the ATP binding site, while reversible inhibitors bind to a specific pocket in the SH3 domain through weak, reversible interactions (e.g., hydrogen bonds or hydrophobic interactions), causing an inactive conformation of the enzyme [6]. The first-in-class BTKi was ibrutinib, which was approved by the Food and Drug Administration (FDA) in 2013. Other approved second- and third-generation BTKis include acalabrutinib, zanabrutinib, and pirtobrutinib (Table 1).

### 2.3. Adverse Effects and Off-Target Activity

The toxicity profile of BTKis is best represented by their pattern of kinase binding, which is mediated by the target inhibition of BTK and variable off-target inhibition. Off-target inhibition includes but is not limited to interleukin-2 inducible T-cell kinase (ITK), tyrosine-protein kinase (TEC), and endothelial growth factor receptor (EGFR) [7]. Table 2 displays the incidence of common adverse effects of FDA-approved BTK inhibitors.

## 3. Hematological Malignancies

### 3.1. Chronic Lymphocytic Leukemia and Small Lymphocytic Lymphoma

Chronic lymphocytic leukemia (CLL) and small lymphocytic lymphoma (SLL) are diseases in which there is an accumulation of leukemic cells that occurs progressively in the peripheral blood, bone marrow, and lymphoid tissues. In 2023, it was estimated that 18,740 people would be diagnosed with CLL in the United States, and an estimated 4490 people would die from the disease, making it the most prevalent adult leukemia in Western countries. CLL and SLL are grouped together due to the majority of similarities shared among both disease states; however, the distinction arises in terms of their manifestations. CLL displays as a significant number of abnormal lymphocytes found circulating in the blood in addition to being resident in bone marrow and lymphoid tissue, whereas in SLL, the majority of disease is in the lymph nodes, bone marrow, and other lymphoid tissues. A diagnosis of SLL is characterized by either very few or no abnormal lymphocytes circulating in the blood. BTK plays a crucial role in the proliferation and survival of white blood cells associated with a diagnosis of CLL/SLL. Ibrutinib, zanubrutinib, acalabrutinib, and pirtobrutinib are the FDA-approved BTKis for use in patients with CLL/SLL.

#### 3.1.1. Ibrutinib

Ibrutinib was the first BTKi to be FDA-approved in 2013 for the treatment of CLL and SLL. Prior to the introduction of BTKis, traditional chemotherapy such as chlorambucil was the standard of care for patients older than 65 years with CLL or SLL. Preliminary results from the RESONATE-2 trial demonstrated the superior safety and efficacy of ibrutinib vs. chlorambucil and supported the initial approval in the United States and Europe. In the updated results of the RESONATE-2 study, after a median follow-up of 82.7 months (range 0.1–96.6 months), significant progression-free survival (PFS) benefit was sustained for ibrutinib vs. chlorambucil (hazard ratio [HR], 0.154; 95% confidence interval [CI] 0.108–0.220) in patients 65 years or older with previously untreated CLL without del(17p). Ibrutinib also improved PFS compared to chlorambucil in patients with high-risk genomic features: del(11q) (HR, 0.033; 95% CI, 0.010–0.107) or unmutated immunoglobulin heavy chain variable region (HR, 0.112; 95% CI, 0.065–0.192). The overall survival (OS) at seven years was 78% with ibrutinib [8]. The Alliance North American Intergroup Study (A041202) compared bendamustine plus rituximab vs. ibrutinib (Arm 2) vs. ibrutinib plus rituximab (Arm 3) to determine whether ibrutinib containing regimens are superior to chemoimmunotherapy (CIT) in terms of PFS. This study showed primary benefit for ibrutinib monotherapy and ibrutinib plus rituximab in patients with unmutated immunoglobulin heavy chain variable region genes (IGHV; 61% of study patients had unmutated IGHV) rather than mutated IGHV. The presence of complex karyotypes did not impact PFS among patients treated with ibrutinib alone [9]. The E1912 study and the FLAIR study showed that ibrutinib + rituximab was more effective than FCR for patients 70 years or less without del(17p)/TP53 mutation, especially for those with unmutated IGHV, indicating that ibrutinib may be an appropriate option for younger patients with IGHV unmutated CLL as well [10]. Based on the results of these trials, ibrutinib monotherapy was approved as a first-line option for patients with CLL/SLL.

#### 3.1.2. Acalabrutinib

Acalabrutinib is currently FDA-approved for the treatment of both untreated and relapsed/refractory CLL based on the results of the ELEVATE-TN trial. In this multicenter phase III study, patients (N = 535) aged 65 years or older, or 18–65 years with comorbidities were randomized (1:1:1) to acalabrutinib monotherapy (n = 179) or acalabrutinib plus obinutuzumab (n = 179) or obinutuzumab plus chlorambucil (n = 177). In the obinutuzumab plus chlorambucil arm, there were 69 patients (39.0%) who crossed over to acalabrutinib. The median PFS was not reached. The 48-month PFS rates were 74.8% and 76.2%, respectively, for acalabrutinib plus obinutuzumab and acalabrutinib monotherapy in patients with del(17p) and/or TP53 mutation [11].

#### 3.1.3. Zanubrutinib

Zanubrutinib, a next-generation BTKi, previously approved in the relapsed/refractory setting, is now indicated for patients with untreated CLL based on the results of the phase III SEQUOIA study. Patients (N = 479) with treatment-naïve CLL/SLL without del(17p) were randomized to receive zanubrutinib (n = 241) or bendamustine plus rituximab (n = 238). Patients in the zanubrutinib arm had a higher ORR (95% vs. 85%) and PFS compared to patients in the bendamustine plus rituximab arm (HR 0.42; *p* < 0.0001) and a suitable safety profile. Statistically significant PFS was also observed in patients with del(11q) and unmutated IGHV (HR, 0.24; *p* < 0.0001), but not for patients with mutated IGHV (HR, 0.67; *p* = 0.0929) [12].

#### 3.1.4. Pirtobrutinib

Patients with lymphoproliferative disorders that are resistant to covalent BTK inhibitors have an unmet therapeutic need. The most recently approved BTKi, pirtobrutinib, is a noncovalent BTKi that has been shown to achieve increased response rates in patients who are refractory to currently FDA-approved non-covalent BTKi. In the BRUIN study, patients (N = 247) with previously treated CLL (refractory to covalent BTKis), were enrolled to receive pirtobrutinib. This study demonstrated an ORR of 82%. ORR was 77.8%, (median PFS, 16.8 months) in patients who were penta-refractory (previous cBTKi, chemotherapy, CD20 antibody, BCL2i, and PI3Ki). The median PFS in the whole cohort was 19.4 months (16.8 months for double-refractory) [13]. Pirtobrutinib is currently indicated in CLL/SLL as a second- or third-line in cases of resistance or intolerance to prior covalent BTKi therapy.

### 3.2. Non-Hodgkin Lymphoma

Non-Hodgkin lymphomas (NHL) are a heterogeneous group of lymphoproliferative disorders originating in B-lymphocytes, T-lymphocytes, or natural killer (NK) cells. In 2023, an estimated 80,550 people will be diagnosed with NHL and there will be approximately 20,180 deaths due to the disease [14]. BTKi has a role in the management of several subtypes of NHL that we will examine here.

#### 3.2.1. Mantle Cell Lymphoma

Mantle cell lymphoma (MCL) typically consists of a more aggressive disease course, as it possesses unfavorable characteristics of both indolent and aggressive NHL. MCL is characterized by the reciprocal chromosomal translocation t(11;14), juxtaposing the cyclin D1 locus with the immunoglobulin heavy chain (IGH) gene locus, leading to overexpression of cyclin D1 [15]. Ibrutinib, zanubrutinib, acalabrutinib, and pirtobrutinib are BTKi FDA-approved for use in patients with MCL.

##### Ibrutinib

The combination of ibrutinib and rituximab demonstrated durable remissions in patients with relapsed or refractory MCL in the four-year follow-up results of a multicenter, phase II clinical trial. At a median follow-up of 47 months (range 1–52 months), 29 (58%) patients had achieved complete remission. The median PFS was 43 months (range 1–48 months) and the three-year PFS was 54%. The median OS was not reached, and the three-year OS was 69% [16].

##### Acalabrutinib

In the single-arm, phase II ACE-LY-004 study, patients (N = 124) with relapsed or refractory MCL were treated with acalabrutinib monotherapy. With a median follow-up of 15.2 months, 100 (81%) patients achieved an overall response, and 49 (40%) patients achieved a complete response. The 12-month PFS rate was 67%, and the OS rate was 87%. A long-term follow-up (>24 months) confirmed these initial findings, with the median PFS being 20 months and the estimated 24-month OS rate being 72% [17].

##### Zanubrutinib

Based on the results of a multicenter, single-arm study, zanubrutinib gained FDA approval for the treatment of relapsed or refractory MCL after at least one prior therapy. In the phase II study, patients (n = 86) with relapsed or refractory MCL were treated with zanubrutinib. After a median follow-up of 18.4 months, 72 (84%) patients achieved an objective response, with 59 (68.6%) achieving a complete response (CR) [18].

##### Pirtobrutinib

The approval of pirtobrutinib in the setting of MCL was based on findings in the BRUIN trial. Patients (N = 90) previously treated for MCL (median of three prior lines of therapy) with prior BTKi treatment received oral pirtobrutinib until disease progression or unacceptable toxicity. At a median follow-up time of 12 months, the median DOR among the 52 responding patients was 22 months. Patients who discontinued their prior BTKi due to disease progression (n = 74) had an ORR of 50% and a median DOR of 14.8 months [19].

#### 3.2.2. Marginal Zone Lymphoma

Marginal zone lymphoma (MZL) originates in the marginal zone of lymphoid follicles located in the mucosa-associated lymphoid tissues (MALT), spleen, and lymph nodes. MZL can be subcategorized into gastric MALT lymphoma, non-gastric MALT lymphoma, nodal MZL, and splenic MZL [20]. Ibrutinib and zanubrutinib are the two FDA-approved BTKis preferred for the treatment of relapsed/refractory MZL in fit as well as elderly or unfit patients.

##### Ibrutinib

In the multicenter phase II PCYC-1121 study the safety and efficacy of single-agent ibrutinib in patients (N = 63) with relapsed/refractory MZL treated with prior rituximab (RTX) or rituximab-based chemoimmunotherapy (RTX-CIT). The final analysis was reported with a median follow-up of 33.1 months (range 1.4–44.6 months). The overall response rate (ORR) was 58%, and the median PFS was 15.7 months (95% CI 12.2–30.4). median OS was not reached. Patients with prior RTX treatment had better outcomes vs. those with prior RTX-CIT treatment (ORR 81% vs. 51%). The ORR results were 63%, 47%, and 62% for extranodal, nodal, and splenic subtypes, respectively [21].

##### Zanubrutinib

Zanubrutinib was FDA-approved for the treatment of relapsed or refractory MZL based on the results of the MAGNOLIA trial. In this multicenter phase II trial patients (N = 68) with relapsed/refractory MZL after ≥1 line of therapy (including ≥1 anti-CD20-monoclonal antibody-based regimen). With a median follow-up of 15.7 months (range 1.6–21.9 months), the ORR for zanubrutinib was 68.2%, and the complete response (CR) was 25.8% among 65 patients evaluable for efficacy. The median PFS was not reached. The duration of response (DOR) at 12 months was 93.0% and PFS was 82.5% at both 12 and 15 months. The ORR results were 64%, 76%, 67%, and 50% in extranodal, nodal, splenic, and indeterminate subtypes, respectively [22].

#### 3.2.3. Waldenstrom’s Macroglobulinemia

Waldenstrom’s macroglobulinemia (WM) is a rare hematological malignancy characterized by the infiltration of lymphoplasmacytic lymphoma into the bone marrow, often accompanied by IgM monoclonal gammopathy. This condition predominantly affects older individuals with a median diagnosis age of 71 years, making many of these patients unsuitable candidates for chemoimmunotherapy. This underscores the urgent need for research to develop safe and effective therapies for WM.

##### Ibrutinib

In January 2015, the BTKi ibrutinib gained approval for the treatment of WM based on a phase II multicenter study involving 63 previously treated WM patients [23]. This study reported a remarkable ORR of 90.5%, with a 69.1% PFS and 95.2% OS at the two-year mark. Building upon this success, Dimopoulos et al. explored the combination of ibrutinib with rituximab in treatment-naïve and recurrent WM patients [24]. The outcomes were promising, with a 30-month PFS of 82% in the ibrutinib–rituximab group compared to 28% in the rituximab-only group, translating to an 80% reduction in the risk of progression or death in the ibrutinib–rituximab group. However, subgroup analysis revealed that patients with MYD88L265P/CXCR4WHIM and MYD88WT disease had lower response rates [24,25]. Moreover, ibrutinib’s multiple off-target activities were associated with various moderate adverse events, including atrial fibrillation and bleeding events. In response to these challenges, a second-generation BTKi with minimal off-target activity, zanubrutinib, was initially evaluated in a phase I/II study for WM patients who were treatment-naïve or had relapsed/refractory disease [26]. This was especially considered for patients with resistant disease (MYD88L265P/CXCR4WHIM and MYD88WT) or those experiencing toxicity. The study enrolled 77 patients (24 treatment-naïve and 53 relapsed/refractory) from 2014 to 2018, and the results showed that long-term treatment with zanubrutinib was well tolerated and led to deep and durable responses in most patients regardless of their mutation status. Notably, the toxicity profile of zanubrutinib was similar to that of the pivotal ibrutinib study but showed significantly less diarrhea (19.5% vs. 42%). More recently, the ASPEN trial, a phase III randomized trial, was designed to directly compare the safety and efficacy of ibrutinib and zanubrutinib in WM patients. This study demonstrated a higher frequency of very good partial response (VGPR) in the zanubrutinib arm after a median follow-up of 19.4 months. In terms of safety, there was an increased risk of developing atrial fibrillation and hypertension with ibrutinib compared to zanubrutinib [27].

Furthermore, other BTKis such as acalabrutinib and tirabrutinib have also been investigated for WM treatment. Both are highly selective second-generation BTKis with minimal off-target activity. A single-arm, multicenter, phase II trial recruited 106 patients with treatment-naïve and relapsed/refractory WM to assess the activity and safety of acalabrutinib [28]. At a median follow-up of 27.4 months, 93% of treatment-naïve patients and 93% of relapsed/refractory patients achieved an overall response, although it is worth noting that some patients discontinued treatment. Tirabrutinib was evaluated in 27 WM patients, with an ORR of 94% and 100% in treatment-naïve and previously treated patients, respectively [29].

## 4. Autoimmune Disorders

### 4.1. Rheumatoid Arthritis

Rheumatoid arthritis (RA) is a chronic inflammatory disease characterized by persistent joint inflammation. In recent times, the therapeutic armamentarium for RA has grown exponentially with the development of biological therapies like abatacept, adalimumab, anakinra, etanercept, infliximab, and rituximab, which has led to improved outcomes significantly. However, there remains an unmet need for patients who do not respond sufficiently or are refractory to current therapies. More therapeutic targets are being investigated, including BTKis. RA was the most frequent indication investigated for BTKis from autoimmune disorders.

Fenebrutinib, an oral non-covalent and highly selective BTKi was investigated in combination with methotrexate (MTX) for patients with RA and inadequate response to MTX or tumor necrosis factor (TNF) inhibitors [30]. In this study, patients in cohort 1 were randomly assigned to different treatment groups in a 1:1:1:1:1 ratio. These groups included oral fenebrutinib at varying doses (50 mg once daily, 150 mg once daily, or 200 mg twice daily), 40 mg adalimumab given by subcutaneous injection every other week, or a placebo. After 40 patients were assigned to each of the five treatment groups, randomization to the fenebrutinib 50 mg once daily group was stopped because it was unlikely to be effective. The remaining patients were then randomly assigned equally to the other treatment groups. In cohort 2, patients were randomly assigned in a 1:1 ratio to either receive a placebo or the highest dose of fenebrutinib evaluated in cohort 1, which was 200 mg twice daily. Cohort 2 patients were expected to have more resistant disease compared to cohort 1. The study lasted for 12 weeks, and patients who completed it had the option to participate in a long-term open-label extension study, where all patients received fenebrutinib at a dose of 200 mg twice daily. Patients in cohort 1 achieved the American College of Rheumatology (ACR) response criteria of 50% at 12 weeks in patients receiving 150 mg daily and 200 mg twice daily (28% and 35% respectively) compared to 15% of the placebo group. It is noteworthy that fenebrutinib 200 mg twice daily had a similar response rate to adalimumab (36%) at 12 weeks.

On the other hand, spebrutinib was explored in a phase IIa multicenter in females with RA who were receiving stable methotrexate doses as background therapy to compare the efficacy, safety, and pharmacodynamic properties of spebrutinib. The study enrolled 47 patients and achieved a 41.7% ACR20 response at four weeks vs. 21.7% placebo patients, but did not meet the primary endpoint or secondary clinical endpoints. It was noted that the low recruitment and short follow-up time along with the dosing used may have underscored the true potential of ibrutinib in RA.

### 4.2. Systemic Lupus Erythematosus

BTKis have emerged as a potential therapeutic avenue in systemic lupus erythematosus (SLE), prompting extensive exploration for new treatment strategies. Numerous preclinical trials have substantiated the efficacy of BTKis in mouse models of lupus. One study showcased the effectiveness of a potent BTKi (PF-06250112) in dose-dependently reducing anti-dsDNA levels, preventing proteinuria development, and ameliorating glomerular pathology scores across all treated groups [31]. Another study demonstrated that early treatment with a different BTKi (BI-BTK-1) forestalled proteinuria development, correlating with significant renal histological protection, reduced anti-DNA titers, and improved OS in both strains [32].

In the realm of clinical trials, up to four BTKis (branebrutinib—NCT04186871, eslubrutinib—NCT03978520, evobrutinib—NCT02975336, and fenebrutinib—NCT02908100) have been investigated, with results available for two of them. While these studies did not reveal any concerning safety events, regrettably, neither of them met their predefined primary endpoints [33,34].

### 4.3. Multiple Sclerosis

The advancement of disease-modifying therapies has significantly transformed the landscape of Multiple Sclerosis (MS), which is evident based on the cost savings secondary to the shift from an inpatient to an outpatient setting [35]. However, substantial opportunities for improvement persist, particularly in the setting of relapsing-remitting multiple sclerosis (RMS) and secondary-progressive multiple sclerosis (PMS).

Within the landscape of emerging therapies, BTKis have recently emerged as a promising frontier. The safety and efficacy of five BTKis are currently under evaluation for MS treatment. For instance, evobrutinib is under evaluation in a phase II trial involving 267 patients randomly assigned to one of five trial groups: placebo, evobrutinib at different doses (25 mg once daily, 75 mg once daily, or 75 mg twice daily), and open-label dimethyl fumarate (DMF) for reference [36]. The primary endpoint was the cumulative number of gadolinium-enhancing lesions identified on T1-weighted magnetic resonance imaging (MRI) at weeks 12, 16, 20, and 24. The trial noted a reduction in the total number of enhancing MRI lesions with evobrutinib at a dose of 75 mg once daily when compared to the placebo between weeks 12 and 24. However, treatment with any dose of evobrutinib did not impact the annualized relapse rate or disability progression and was associated with elevations in liver aminotransferase levels.

Another BTKi, tolebrutinib, was investigated in a multicenter, 16-week, phase IIb, randomized, double-blind, placebo-controlled, crossover, dose-finding trial for patients with relapsing MS [37]. Eligible participants underwent a two-step randomization process, with subsequent assignment to four tolebrutinib dose groups (5 mg, 15 mg, 30 mg, and 60 mg) administered once daily as an oral tablet. The primary efficacy endpoint was the number of new gadolinium-enhancing lesions detected after 12 weeks of tolebrutinib treatment compared to lesions accumulated during a four-week placebo run-in period. This trial demonstrated a dose-related reduction in the number of new enhancing lesions after 12 weeks of tolebrutinib treatment while being well-tolerated.

## 5. Solid Tumors

Preclinical data indicates the potential role of BTKis in regulating the tumor microenvironment of specific tumor subtypes, involving cells such as dendritic cells, macrophages, myeloid-derived suppressor cells, and endothelial cells [38]. The exploration of ibrutinib has extended across various solid malignancies, including lung, breast, pancreatic, colon, and neuroendocrine cancers.

Hong et al. conducted a prospective, multicenter, open-label, phase Ib/II study to assess the safety, tolerability, and preliminary effectiveness of combining ibrutinib with a PD-L1 inhibitor (durvalumab) in patients with advanced solid tumors. The study enrolled 124 patients, comprising 50 with pancreatic adenocarcinoma, 45 with breast cancer, and 29 with non-small cell lung cancer, predominantly at stage IV (94%) and with a median of three prior lines of therapy. The combination of 560 mg ibrutinib daily and 10 mg/kg durvalumab every two weeks was deemed safe, with the most common treatment-related adverse events including fatigue (43%), nausea (28%), decreased appetite (26%), hypomagnesemia (23%), anemia (21%), peripheral edema (21%), diarrhea (21%), and dyspnea (21%). This dosage combination was established as the recommended phase II dose. However, only two patients in the overall study population achieved a confirmed response, and six achieved stable disease. The median OS for the entire study population was six months, with variations noted in different cancer cohorts [39].

The RESOLVE study, a phase III, randomized, double-blind, placebo-controlled trial, involved 424 patients receiving once-daily oral ibrutinib (560 mg) or placebo alongside nab-paclitaxel and gemcitabine. After a median follow-up of 25 months, no significant difference in OS was observed between the treatment groups [40].

In the study by Kim et al., the combination of ibrutinib with pembrolizumab in patients with refractory metastatic proficient mismatch repair colon cancer was investigated. No dose-limiting toxicity was observed, and while stable disease was achieved in some patients, no objective response was recorded. The median PFS and OS were 1.4 and 6.6 months, respectively [41].

Additionally, a prospective phase II trial explored ibrutinib’s use in patients with advanced gastrointestinal/lung neuroendocrine neoplasms (NENs) and pancreatic NENs who had evidence of progression within 12 months of study entry on at least one prior therapy. However, no objective responses were noted, with a median progression-free survival of three months and several drug-related adverse events associated with ibrutinib [42].

## 6. Graft-versus-Host Disease

Allogeneic hematopoietic stem cell transplant (HSCT) has long held significant curative potential, albeit with reservations due to its potential adverse effects. In fact, up to 50% of patients who undergo HSCT from matched sibling donors are susceptible to graft versus host disease (GVHD), a figure that can soar to 70% in unrelated donor transplants [43]. Fortunately, recent advances in GVHD management, including post-transplant cyclophosphamide (PTCy), in-vivo T-lymphocyte cell depletion [44,45,46], and ex-vivo T-cell depletion [47,48,49], have improved outcomes and expanded the use of haploidentical HSCT. Nevertheless, despite progress in preventing acute GVHD, chronic GVHD (cGVHD) remains a significant source of morbidity, underscoring the urgent need for research into steroid-sparing therapies.

The pathophysiology of cGVHD implicates both T- and B-cells. While T-cell depletion has shown promise in reducing incidence, it comes at the cost of heightened infection risk and disease relapse [50,51]. Additionally, B-cell activation of the BTK pathway promotes B-cell survival [52], while ITK contributes to the activation of T-cells against healthy tissues [53]. Notably, ibrutinib has demonstrated efficacy against both cell types in preclinical trials. In 2014, Miklos et al. initiated a multicenter, open-label study to evaluate the safety and effectiveness of ibrutinib in patients with active cGVHD who had not responded adequately to steroid-based therapies. Remarkably, they achieved a remarkable ORR of 67% among 42 patients, reducing the need for steroids with an acceptable safety profile [54]. An extended follow-up in 2019 reported even better outcomes, with an increased complete response rate, sustained responses over time, and minimal steroid doses [55].

More recently, the phase I/II iMAGINE trial explored ibrutinib’s use in the pediatric population, establishing recommended pediatric equivalent doses and assessing pharmacokinetics and safety in treatment-naïve or relapsed/refractory moderate to severe cGVHD patients. As of its publication in November 2022, the results surpassed those in adults, demonstrating not only durable responses but also greater response rates [56]. Conversely, the iNTEGRATE study, a randomized, double-blind, placebo-controlled, phase III trial, evaluated the safety and efficacy of Ibrutinib in combination with prednisone for previously untreated cGVHD patients. While primary endpoints were not met, other indicators such as immunosuppression withdrawal and patient-reported outcomes presented encouraging results [57]. Future studies are warranted to further explore the advantages of ibrutinib over steroid-based therapies.

## 7. Future Directions

As discussed, BTKis have a wide range of clinical practice in several benign and malignant hematological disorders with growing application in other auto-immune disorders. However, overcoming resistance to BTKis remains a major issue. The three commonly used BTKis, namely ibrutinib, acalabrutinib, and zanubrutinib, work with covalent and irreversible binding with the downstream BTK domain. One of the common mechanisms of resistance to the covalent BTKis is through the C481 mutation. The non-covalent inhibitors offer hope for bypassing this resistance mechanism. Pirtobrutinib is the only non-covalent inhibitor with regulatory approval in the USA at this time for MCL in relapsed/refractory settings. In a phase I/II trial, it was shown to have an ORR of 73% in a total of 317 patients with CLL/SLL 247 of these patients had prior exposure to a BTK [58]. The ongoing phase III LOXO-BTK-20019 trial (NCT04662255) is comparing its efficacy to the three traditional BTKis in the BTK naïve population in patients with mantle cell lymphoma with at least one prior therapy. Nemtabrutinib (MK-1026) is another non-covalent BTKi that has shown activity against the mutated C481S BTK in addition to the wild-type. In the BELLWAVE-001 study of 112 patients with heavily pretreated hematological malignancies, 57 had CLL/SLL and among those 63% with C481S mutation 32% harbored a p53 mutation and 58% had IGHV-unmutated disease. An ORR of 56% was seen in the CLL/SLL subgroup [59].

Another novel and promising class of drugs is BTK degraders. Traditional BTKis work by inhibiting the activity of the enzyme, which can lead to resistance mechanisms including kinase-dead mutations where the enzymatic activity virtually ceases. However, these mutations lead to the onset of different interactions with the end result of ongoing B-cell receptor signaling. The BTK degraders are a class of heterobifunctional molecules known as proteolysis-targeting chimeras (PROTAC). These promising drugs are composed of two active domains and a linker, with one of the active domains (BTK ligand in this case), shuttling the kinase towards an E3 ubiquitin ligase. This enables the interaction of BTK degraders with wild and mutant forms of the kinase. NX-2127 is a PROTAC that targets IKZF1 and IKZF3 transcription factors in addition to BTK and is showing clinical activity in patients who had been treated with other BTKis including pirtobrutinib [60]. An ongoing phase Ia/Ib trial (NCT04830137) is exploring its use in R/R B-cell malignancies.

## 8. Conclusions

BTKis were initially heavily investigated in the management of CLL/SLL; however, over the recent years, their use has expanded beyond malignant hematological disorders. It is important to recognize that BTKis are not drugs free of adverse events, especially when taken over an extended period of time. However, they are generally well-tolerated. There is promising data for BTKi use in disorders outside of hematologic malignancies, including autoimmune disorders as well as GVHD.

## Figures and Tables

**Figure 1 ijms-25-02208-f001:**
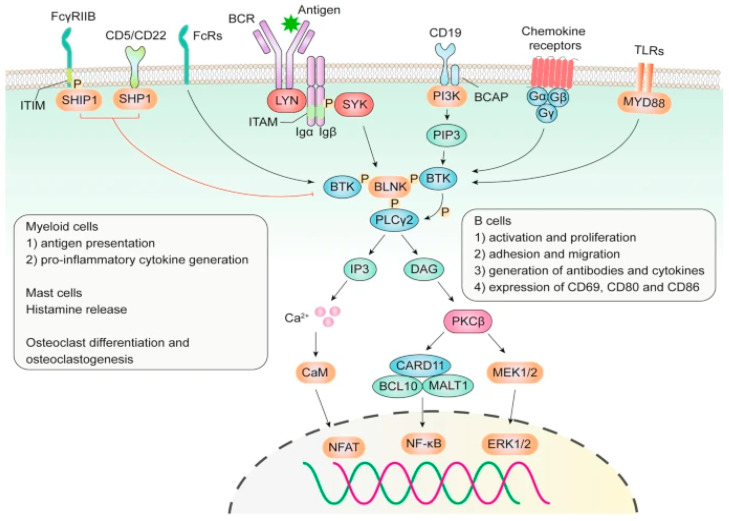
Overview of BTK signaling pathway highlighting the downstream signaling cascade and intracellular signaling pathway blockade [3].

**Table 1 ijms-25-02208-t001:** Summary of approved BTK inhibitors.

BTK Inhibitor	Year of FDA Approval	Generation	BTK Inhibition	Indication
Ibrutinib (Imbruvica)	2013	First	Irreversible	CLL/SLL, GVHD, MCL, MZL, WM
Acalabrutinib (Calquence)	2017	Second	Irreversible	CLL/SLL, MCL
Zanubrutinib (Brukinsa)	2019	Second	Irreversible	CLL/SLL, MCL, MZL, WM
Pirtobrutinib (Jaypirca)	2023	Third	Reversible	CLL/SLL, MCL

Abbreviations: chronic lymphocytic leukemia (CLL); small lymphocytic lymphoma (SLL), graft-versus-host disease (GVHD); mantle cell lymphoma (MCL); marginal zone lymphoma (MZL); Waldenstrom macroglobulinemia (WM).

**Table 2 ijms-25-02208-t002:** Common Adverse Effects of BTK Inhibitors.

Adverse Effect	Ibrutinib	Acalabrutinib	Zanubrutinib	Pirtobrutinib
Arthalgias	16% to 24%	8% to 16%	-	≤12%
Atrial fibrillation	≤8%	≤5%	≤5%	-
Bleeding events	48%	8% to 20%	24% to 42%	11%
Diarrhea	28% to 59%	18% to 35%	14% to 22%	19%
Headache	12% to 21%	22% to 39%	8% to 18%	-
Hypertension	11% to 16%	3%	14% to 19%	-
Infections	10% to 11%	56% to 65%	-	-
Lymphocytosis	66%	16% to 26%	24%	34%
Neutropenia	≥30%	23% to 48%	-	-
Rash	12% to 25%	9% to 25%	20% to 29%	14%

## Data Availability

Not applicable.

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
