# Peer review of "Bruton’s Tyrosine Kinase Inhibitors: Recent Updates"

_ijms, 2024, doi:10.3390/ijms25042208_

Round 1
Reviewer 1 Report
Comments and Suggestions for Authors
The article provides a comprehensive overview of the impact of Bruton’s tyrosine kinase (BTK) inhibitors on diverse medical conditions, from hematological malignancies to autoimmune disorders. It adeptly traces the evolution of BTK inhibitors (BTKi) from their primary focus on managing CLL/SLL to their diversified applications in different disorders. The promising findings in autoimmune disorders and graft-versus-host disease (GVHD) contribute to expanding the scope of BTKi applications, establishing them as versatile therapeutic agents. The proposed review is both timely and clinically significant. Addressing the provided suggestions will enhance the manuscript's overall quality.
1) The inclusion of a pictorial representation illustrating the BTK structure is crucial for enhancing the visual understanding of the readers.
2) It is recommended to tabulate the survival and response rates of BTK inhibitors (BTKi) for various diseases. This tabular format will facilitate a clearer comprehension of the data for the readers.
3) The numbering system for the inhibitors lacks consistency and should be revised to ensure accuracy and clarity throughout the manuscript.
Author Response
1) The inclusion of a pictorial representation illustrating the BTK structure is crucial for enhancing the visual understanding of the readers.
-We have included a pictorial.
2) It is recommended to tabulate the survival and response rates of BTK inhibitors (BTKi) for various diseases. This tabular format will facilitate a clearer comprehension of the data for the readers.
-Survival rate pertaining to each of the disease states and the corresponding BTKi are included in each section with evidence-based data.
3) The numbering system for the inhibitors lacks consistency and should be revised to ensure accuracy and clarity throughout the manuscript.
-We have corrected the order of the drugs to be consistent throughout the article
Thank you for your time!
Reviewer 2 Report
Comments and Suggestions for Authors
The review is about Bruton’s tyrosine kinase (BTK) inhibitors that revolutionized the landscape for treatment of hematological malignancies solid tumors and recently autoimmune disorders. In general, the paper exhibits a commendable level of writing proficiency, and the examination of BTK inhibitors is intellectually stimulating. Table 1 constitutes a significant inclusion that has the potential to enhance the paper's citation count. I endorse the submission of this paper for publication after addressing some of my concerns:
The authors' inclusion of fundamental concepts in introduction section is missing and it should contribute to the extent body of literature.
The objective of the review is not clearly stated in the abstract section.
Introduction:
- Introduction should be covered the gap of the research. However, it is not well covered in this section.
- Also, please mention the important of this study to society as well as industry.
- Problem statement of your introduction is not strong, need to discuss more about it.
- A good introduction should conclude the introduction by mentioning the specific objectives of the research.
- The earlier paragraphs should lead logically to specific objectives of the study.
- Revised Introduction section based on the structure below:
1st paragraph: Problem statement
2nd paragraph: Current ongoing solution
3rd paragraph: Proposed solution in this work.
4th paragraph: Summarized the current research novelty and objective of this work.
- Kindly refer some latest papers as it is highly relevant to this report.
To be honest, there are many reviews about BTK inhibitors. What is the highlight of this manuscript? I think the authors should emphasize and illustrate their own idea on this concept at least in the abstract/conclusion section.
Author should clarify the differences between this review and many previous similar reviews about this topic.
The review is very weak, and the information is scattered around. There is only one table in the manuscript. The authors may add some interactive figures that can help the readers to grasp the overall direction of the paper.
A more systematic review of recent literature would be helpful to have a more in-depth analyses on this topic.
Review is lack of detailed summary of the advantages and disadvantages of various methods helped to identify BTK inhibitors.
Author Response
The aim of our review is to provided updated evidence in support of BTKis and highlight some indications the audience may not be as familiar with, Prior to that we do provide a background of data we are already aware of. We have updated the introduction and provided an objective as you recommended. Each section reviews a indication of BTKis and data supporting each of these indications is relayed in the appropriate section.
Round 2
Reviewer 2 Report
Comments and Suggestions for Authors
No further comments